# Improved Leakage Behavior at High Temperature via Engineering of Ferroelectric Sandwich Structures

**DOI:** 10.3390/ma16020712

**Published:** 2023-01-11

**Authors:** Guangliang Hu, Yinchang Shen, Qiaolan Fan, Wanli Zhao, Tongyu Liu, Chunrui Ma, Chun-Lin Jia, Ming Liu

**Affiliations:** 1School of Microelectronics, Xi’an Jiaotong University, Xi’an 710049, China; 2Science and Technology on Electro-Optical Information Security Control Laboratory, Tianjin 300308, China; 3State Key Laboratory for Mechanical Behavior of Materials, Xi’an Jiaotong University, Xi’an 710049, China

**Keywords:** multilayer structure, ferroelectrics, energy storage and conversion, electrical properties

## Abstract

The leakage behavior of ferroelectric film has an important effect on energy storage characteristics. Understanding and controlling the leakage mechanism of ferroelectric film at different temperatures can effectively improve its wide-temperature storage performance. Here, the structures of a 1 mol% SiO_2_-doped BaZr_0.35_Ti_0.65_O_3_ (BZTS) layer sandwiched between two undoped BaZr_0.35_Ti_0.65_O_3_ (BZT35) layers was demonstrated, and the leakage mechanism was analyzed compared with BZT35 and BZTS single-layer film. It was found that interface-limited conduction of Schottky (S) emission and the Fowler-Nordheim (F-N) tunneling existing in BZT35 and BZTS films under high temperature and a high electric field are the main source of the increase of leakage current and the decrease of energy storage efficiency at high temperature. Only an ohmic conductive mechanism exists in the whole temperature range of BZT35/BZTS/BZT35(1:1:1) sandwich structure films, indicating that sandwich multilayer films can effectively simulate the occurrence of interface-limited conductive mechanisms and mention the energy storage characteristics under high temperature.

## 1. Introduction

The lead-free dielectric material for energy storage application has progress greatly in recent years, especially for the Ba(Zr_x_Ti_1−x_)O_3_ (BZT)-based systems [1,2,3,4,5]. The large bandgap, high dielectric constant, and low dielectric loss make them an excellent material for energy storage applications, and they have great advantages in high-power and high-energy-capacitor energy storage applications. From the practical point of view, on the one hand, with its application in new energy vehicles, the requirement for high temperature stability is getting higher and higher. On the other hand, with the development of miniaturization of electronic devices, temperature stability is also becoming more and more important. All these require the energy storage conversion unit not only to have high energy storage density and efficiency, but also to have good high temperature stability [6,7]. As we know, thermal energy can aid the transport of the charge carrier and strongly influence the leakage current, and the leakage current determines the breakdown field strength and thus the energy storage performance [8,9]. It is exciting that, with the application of interface engineering in multilayer films, the breakdown field strength can be effectively improved, and the wide-temperature storage characteristics can be improved by constructing multilayer structures reasonably [10,11,12,13,14,15,16]. However, the mechanism by which the structure influences the leakage current at high temperature is unclear.

Therefore, in this work, the leakage current mechanism is analyzed deeply based on BaZr_0.35_Ti_0.65_O_3_ (BZT35), 1 mol% SiO_2_-doped BaZr_0.35_Ti_0.65_O_3_ (BZTS) single-layer film and BZT35/BZTS/BZT35(1:1:1) sandwich structure multilayer film, so as to deeply understand the enhancement mechanism of breakdown strength and wide-temperature stability of multilayer film (reported in our previous work [17]). This provides a theoretical basis for further enhancing the wide-temperature energy storage properties of dielectric films.

## 2. Materials and methods

### 2.1. Materials and Fabrication

The epitaxial BZT35/BZTS/BZT35 sandwich-structure multilayer film, as well as the BZT35 and BZTS single-layer film, were grown on (001) oriented 0.7 wt% Nb doped SrTiO_3_ (Nb:STO) substrates through a radio-frequency (RF) magnetron sputtering system. The grown temperature and atmosphere were 700 °C and an Ar/O_2_ (1:1) mixture with the pres of 0.2 mbar, respectively. When grown, the thin films were in situ annealed at 700 °C under 400 mbar of Ar/O_2_ (1:1) mixture for 15 min, and then cooled down to room temperature at a rate of 10 °C min^−1^. The thickness of both single-layer and multilayer thin films were controlled to about 400 nm, and the thickness ratios of the layers in the sandwich structures multilayer thin film was designed as *t*_BZT35_:*t*_BZTS_:*t*_BZT35_ = 1:1:1. 

### 2.2. Material Characterization

For the measurement of the dielectric properties, the 100 nm-thickness top electrode of Pt with the dimension of 200 μm × 200 μm were deposited using the RF-sputtering technology with a shadow mask. The current-voltage characteristics were measured using an Agilent B2901A precision source.

### 2.3. Main Conduction Mechanisms

Ohmic conduction: this conduction mechanism mainly takes the free electrons in the conduction band and the holes in the valence band as the carrier transmission mechanism, and it shows a linear relationship between the current density and the electric field. The current density can be expressed as [18,19]: (1)J=enμE
where *J*, *e*, *n*, *μ*, *E* are the current density, electron charge, electron density in conduction band, electron mobility, and electric field, respectively.

Space-charge-limited conduction (SCLC): in this conduction, the current through this medium has nothing to do with the conductivity of the medium, but is only determined by the space charge in the medium. The current density can be expressed as [20]: (2)$$J=\frac{9}{8}\varepsilon_{r}\varepsilon_{0}\mu \frac{E^{2}}{d}$$
where *ε*_r_, *ε*_0_, *d* are the relative dielectric constant of the film, permittivity of the free space, and film thickness, respectively.

Poole-Frenkel (P-F) emission: P-F emission is similar to Schottky emission, and is also called the internal Schottky emission sometimes. The current density can be expressed as [19,20]: (3)$$J=AEexp\left[\frac{-\left(\phi_{A}-e\sqrt{eE/\pi \varepsilon_{\mathrm{opt}}\varepsilon_{0}}\right)}{kT}\right]$$
where *A*, ϕ_A_, *ε*_opt_, *k*, *T* are a constant in the trap ionization energy, optical dielectric constant of the film, Boltzmann’s constant, and the temperature, respectively. 

Schottky (S) emission: the difference between the electron affinity of an insulator and the metal work function is the barrier height, and when the electrons in the metal obtain enough energy provided by thermal activation, it will overcome the energy barrier to go to the dielectric. The current density can be expressed as [20,21]: (4)$$J=BT^{2}exp\left[\frac{-\left(\phi_{B}-e\sqrt{eE/4\pi \varepsilon_{\mathrm{opt}}\varepsilon_{0}}\right)}{kT}\right]$$
where *B* and ϕ_B_ are a constant and the Schottky barrier height, respectively. 

Fowler–Nordheim (F-N) tunneling: when the applied electric field is large enough, the energy band of the insulator will be thinner, and the electrons can directly tunnel through the insulation layer to generate current. The current density can be expressed as [19,20,21]: (5)$$J=DE^{2}exp\left(\frac{-N\phi_{D}^{3/2}}{E}\right)$$
where *D* and *N* are constants, and ϕ_D_ is the potential barrier height.

## 3. Results and Discussion

### 3.1. Leakage Current Density

The schematic structure of thin films and capacitors integrated on Nb:STO were shown in the Figure 1. For the sandwich structure multilayer thin film, BZTS with a high breakdown electric field (*E*_b_) was used as core layer and BZT35 was used as the outer layer. In order to unveil the relationship between the leakage behavior and structure, the samples of BZT35, BZTS, and BZT35/BZTS/BZT35(1:1:1) were chosen as model. The leakage current density of them at the temperature range of −100~200 °C was investigated, as shown in Figure 2a–c. It can be seen that the leakage current density *J* increase with increasing the temperature under the same electric field for all samples of BZT35, BZTS, and BZT35/BZTS/BZT35(1:1:1). However, when the temperature exceeds a certain level, it can be seen that *J* increases sharply under a high electric field for BTZ and BZT35 monolayer films. It can be seen more clearly from the Figure 2d that the leakage current density of these three samples is *J*_BZT35/BZTS/BZT35(1:1:1)_ < *J*_BZTS_ < *J*_BZT35_. When the temperature above 160 °C, the *J* of BZT35 film increased by 11.5 times from 1.3 × 10^−5^ A/cm^2^ at 160 °C to 1.5 × 10^−4^ A/cm^2^ at 200 °C, and the *J* of BZTS increases 2.8 times. This is because the SiO_2_ has a large band gap and good insulation, and a small amount of SiO_2_ doping can improve the insulation of the BZT film and reduce the leakage current. In contrast, the *J* of BZT35/BZTS/BZT35(1:1:1) increases only 1.5 times from 1.4 × 10^−6^ A/cm^2^ at 160 °C to 2.1 × 10^−6^ A/cm^2^ at 200 °C. The lowest leakage current density BZT35/BZTS/BZT35(1:1:1) in a wide range of temperature makes it have a higher breakdown electric field, and achieves high energy storage density *W*_re_ at 200 °C in our previous report [17].

### 3.2. Leakage Mechanism under 200 °C

For clear understanding of the leakage mechanism, both bulk-limited conductions of the Ohmic, SCLC, P-F emission and interface-limited conductions of S emission, F-N tunneling are used to analyze the *J-E* curves of BZT35, BZTS, and BZT35/BZTS/BZT35(1:1:1) under positive bias at 200 °C. The linear fitting of ln*J* vs. ln*E* is performed on these three samples separately, as shown in Figure 3a. It can be seen that the slope of the fitting line for the whole curve of BZT35/BZTS/BZT35(1:1:1) is about one, indicating that only Ohmic conduction behavior (according to the Equation (1)) exists in the whole range of the test electric field. The current is determined primarily by the concentration of the electrons that are thermally excited and transmitted from the valence band to the conduction band [18,19]. However, for BZT35 and BZTS single-layer films, the ln*J* vs ln*E* curves can be divided in three parts. The first part (*E* < *E*_1_) has a slope of about one, similar with BZT35/BZTS/BZT35(1:1:1), the Ohmic conduction behavior plays the dominant role. The second part (*E*_1_ < *E* < *E*_2_) of slope = 2 indicates an SCLC behavior (according to the Equation (2)). It should be caused by the low mobility of BZT35 and BZTS, where the transport of charges injected from electrode is not timely enough, and remaining charges formed as the space charges. In the third part (*E* > *E*_2_), the slope is greater than two. The sharp increase in leakage current can be interpreted as the trap being fully charged. As the P-F mechanism dominated by bulk-limited conduction usually occurred at a higher electric field than the SCLC conduction, and in this mechanism the energy barrier height of the carrier jumping to the valence or conductive band decreases so that more carriers can jump over the energy barrier height to the valence or conductive band, we conducted a linear fitting of the ln(*J*/*E*) and *E*^1/2^ relationship curves (according to the Equation (3)) of BZT35 and BZTS films, as shown in Figure 3b. It can be seen that the corresponding optical refractive index *n* calculated from the ln(*J*/*E*) and *E*^1/2^ curves of both BZT35 and BZTS films are much greater than two (*n* is around two for BZT35 and BZTS [22,23,24,25]), indicating that there is no P-F conduction mechanism in these two samples. 

In order to uncover the conduction mechanism of BZT35 and BZTS films in the larger electric filed (the part of *E* > *E*_2_, in Figure 3a), the interface-limited conduction mechanism of S emission and F-N tunneling are explored [18,19,20,21,26]. The linear fitting of ln(*J*) and *E*^1/2^ was investigated first, as shown in Figure 3c. The value of *n*, calculated by the fitting of both BZT35 and BZTS, is about two (insert of Figure 3c). This means the S emission (according to the Equation (4)) plays a key role in both BZT35 and BZTS films. Then the linear fitting of ln(*J*/*E*^2^) vs. (1/*E*) curves (as shown in Figure 3d) are investigated to confirm whether leakage current is dominated by the F-N tunneling or not. It can be seen that both BZT35 and BZTS have a linear fitting at a high electric field (low 1/*E*), indicating the F-N tunneling conductive mechanism (according to the Equation (5)) exists in both BZT35 and BZTS films at a high electric field.

It can be seen from the discussion above of conduction mechanisms at 200 °C for BZT35, BZTS and BZT35/BZTS/BZT35(1:1:1) that the current at a low electric field is derived from the Ohmic conduction behavior, then from SCLC behavior at a higher electric field, while S emission and F-N tunneling conductive mechanisms act at much higher electric field for BZT35 and BZTS films. For BZT35/BZTS/BZT35(1:1:1) sandwich structure film, only Ohmic conduction behavior is founded at the measured electric field range.

### 3.3. Leakage Mechanism under All Temperature

To have a more intuitive understanding of the leakage behavior of BZT35, BZTS, and BZT35/BZTS/BZT35(1:1:1) at different temperature, the conduction mechanism under both positive and negative biases at different temperatures are shown in Figure 4a–c. It can be seen that there is only bulk-limited conduction (Ohmic conduction and SCLC behavior) at low temperature under the electric field of −3.0 MV/cm to 3.0 MV/cm for BZT35 and BZTS films. With the increasing of temperature, it shows interface-limited conduction mechanism of F-N tunneling at a negative bias under a high electric field (120 °C) and then S emission co-exists with F-N tunneling at high temperature (140 °C) for negative bias, 160 °C for positive bias for BZT35 films, as shown in Figure 4a, which causes the energy storage density to decrease sharply, as shown in Figure 4d. For BZTS films, as shown in Figure 4b, the interface-limited conduction mechanism of F-N tunneling at both negative and positive biases under a high electric field (160 °C) and then S emission co-exists with F-N tunneling at high temperature (180 °C). This means that the interface-limiting conduction mechanism is narrower than BZT35 film, and the doped SiO_2_ is beneficial to obtain a relatively lower leakage current. It is exciting that only Ohmic conduction is found in the temperature range from −100 °C to 200 °C under the tested electric field for BZT35/BZTS/BZT35(1:1:1) film, as shown in Figure 4c. The conduction can be described from the insert energy band diagram in Figure 3c: the electrons obtain activation energy Φ, and transition to the conduction band of the dielectric material to form conduction. This indicates that BZT35/BZTS/BZT35(1:1:1) sandwich structures can effectively block the thermal excitation of electrons and prevent the electrons from traps into the conduction band of the dielectric. From the energy storage efficiency *η* under the different temperature of these three samples at 3 MV/cm showed in Figure 4d, it can be seen that the *η* decreases with increasing the temperature for all of them, and the *η* of BZT35 decreases more quickly than BZTS, especially at high temperature. The trend of *η*_BZT35/BZTS/BZT35(1:1:1)_ > *η*_BZTs_ > *η*_BZT35_ (in Figure 4d) and *J*_BZT35/BZTS/BZT35(1:1:1)_ < *J*_BZTS_ < *J*_BZT35_ (in Figure 2d) shows that the *η* is closely related to the leakage current of the film, the higher leakage current, the lower *η*. This proves that the sandwich structure film of BZT35/BZTS/BZT35(1:1:1) successfully reduces the leakage current of the material at high temperature, thus achieving high breakdown field strength and excellent energy storage performance at high temperature.

## 4. Conclusions

In this work, the leakage current of BZT35, BZTS and BZT35/BZTS/BZT35(1:1:1) was investigated and detailed, and the conduction mechanism of them at different temperature under different electric fields are discussed. The S emission and F-N tunneling conduction of interface-limited conduction mechanisms are the main source of the increasing the leakage current of BZT35 and BZTS film at a high electric field and high temperature. This makes them have a lower breakdown field strength and a significant decrease in energy storage efficiency at high temperatures. On the contrary, the leakage behavior of BZT35/BZTS/BZT35(1:1:1) sandwich structures under the whole test electric field and temperature only has an ohmic conduction mechanism, which is the main reason for its high breakdown field strength and good stability at wide temperature.

## Figures and Tables

**Figure 1 materials-16-00712-f001:**
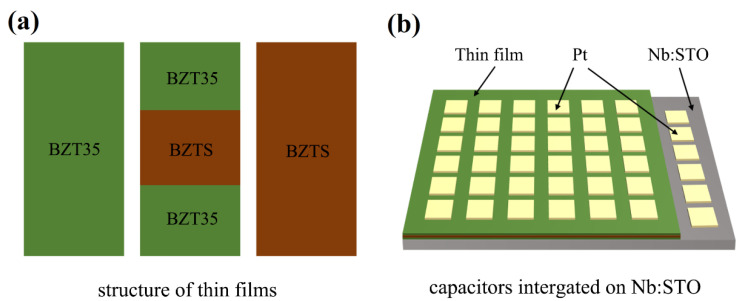
(**a**) Structure of BZT35 and BZTS single-layer and BZT35/BZTS/BZT35(1:1:1) sandwich structure multilayer films. (**b**) Diagram of capacitors integrated on Nb:STO.

**Figure 2 materials-16-00712-f002:**
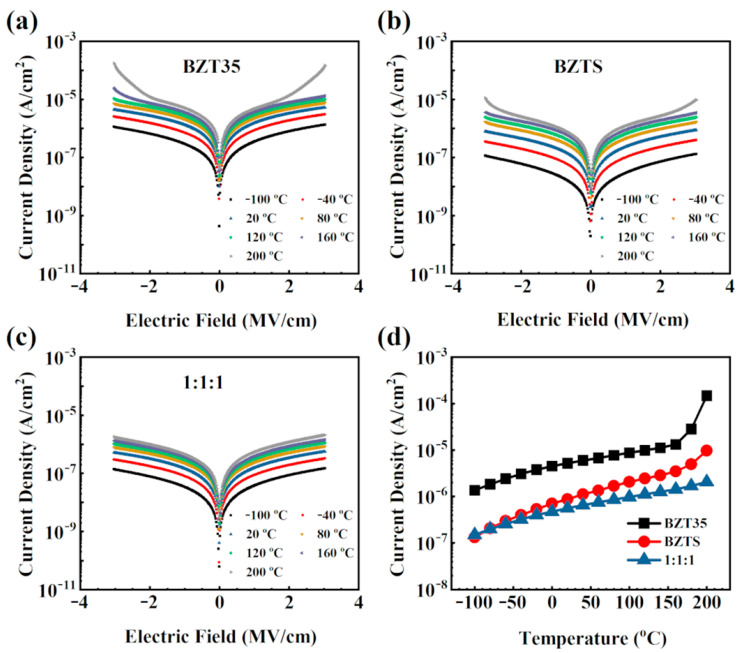
Current density (*J*) vs. electric field (*E*) curves for BZT35 (**a**), BZTS (**b**) and BZT35/BZTS/BZT35(1:1:1) (**c**) films. (**d**) Current density (*J*) at 3.0 MV/cm under different temperature for these three films.

**Figure 3 materials-16-00712-f003:**
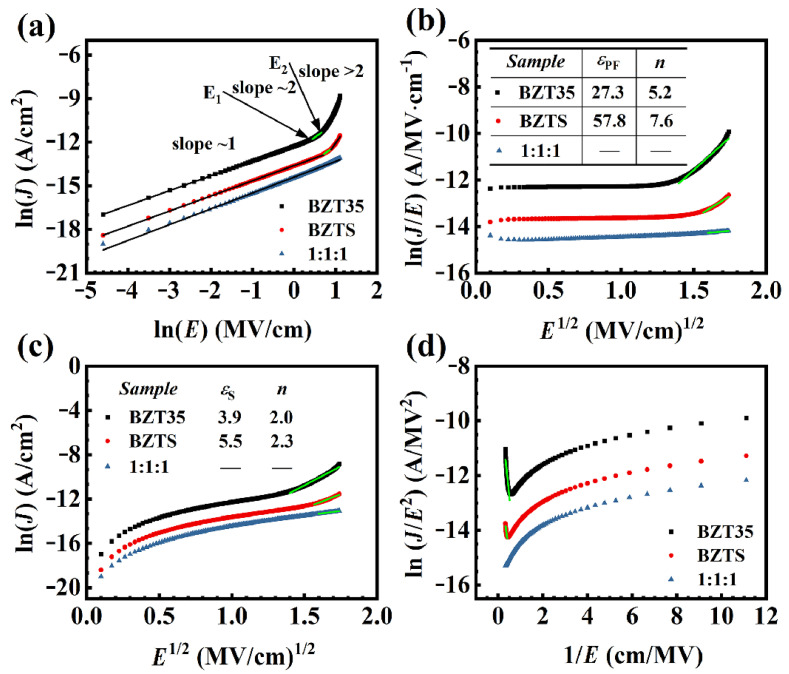
The ln(*J*) vs. ln(*E*) (**a**), ln(*J*/*E*) vs *E*^1/2^ (**b**), ln(*J*) vs *E*^1/2^ (**c**) and ln(*J*/*E*^2^) vs. 1/*E* (**d**) for BZT35, BZTS and BZT35/BZTS/BZT35(1:1:1) films under 200 °C.

**Figure 4 materials-16-00712-f004:**
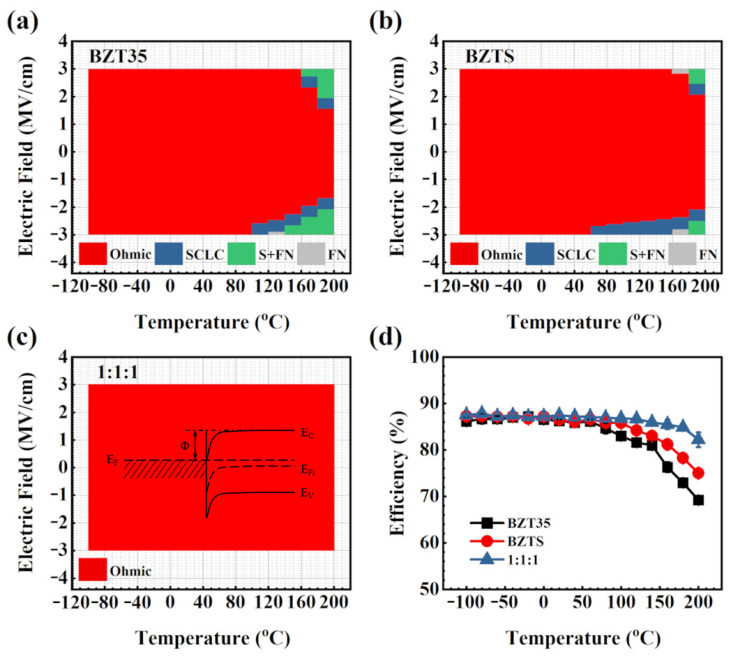
An overview of the leakage mechanisms for BZT35 (**a**), BZTS (**b**) and BZT35/BZTS/BZT35(1:1:1) (**c**) films at positive and negative bias under the temperature from −120 °C to 200 °C, insert is the energy band diagram of ohmic conduction. (**d**) The energy storage efficiency of them under the temperature from −120 °C to 200 °C.

## Data Availability

Data sharing is not applicable to this article.

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
