# Peer review of "Improved Leakage Behavior at High Temperature via Engineering of Ferroelectric Sandwich Structures"

_materials, 2023, doi:10.3390/ma16020712_

Round 1

Reviewer 1 Report

This paper is devoted to the improvement of leakage behavior via engineering of ferroelectric sandwich structures. The authors first consider the conduction mechanisms of the consistuents BZT35 and BZTS layers. After, the show that the insertion of a BZTS layer reduces Schottky emission and Fowler-Nordheim tunneling leaving only ohmic conduction. 

There are some shortages in the desciption of Schottky emission (the electric field lowers the surface barrier by an amount ΔW, increasing the emission current) and Fowler-Nordheim tunneling (FNT, electron tunneling through a triangular shape part of the barrier to the conduction band of the insulator).

Moreover, the following items should be cleared:

1) The interface-limited conduction mechanisms of Schottky emission and FNT depend on barrier height which is obviously modified by insertion of an interface layer. This topic should be discussed by the authors. 

2) Since the sandwich structure obeys ohmic behaviour a band diagram would be useful for illustration.

3) According to figure 2d, the lower leakage of the sandwich structure is attributed to a lower leakage current of BZTS. Here the authors have to give an explanation of the role of the SiO2 fraction in reducing leakage.

4) While Schottky emission is temperature dependent, FNT is a temperature independent tunneling process. Figure 4a and 4b demonstrate the appearance of FNT at elavated temperatures above 120°C. This yields an activation energy of 34 meV, typical for shallow traps. Can you exclude Pool-Frenkel emission in this case ?

Reviewer 2 Report

The paper is interesting study of leakage behavior of ferroelectric SiO2-doped BaZr0.35Ti0.65O3 (BZTS) layer sandwich structures at high temperature. The study of mechanisms of leakage behavior of ferroelectric is very important for improvement of energy storage parameters. Interface in multilayer films effects on breakdown field strength and that is why it is possible to improved storage parameters by interface engineering. But for this it is needed to understand the mechanism  of influence of the structure on the leakage current at high temperature. In the paper both theoretical and experimental study of this mechanism were provided.

The paper is careful, original study and will be interesting for readers. It should be published after some corrections.

1.      The article investigates the phenomena at the ferroelectric interface, taking into account the also radiation component. In this case Maxwell–Wagner–Sillars polarization also should play a role. However, there is no estimation of  the role of Maxwell–Wagner–Sillars polarization in the article.

2.      Also, estimations of  Jonscher dielectric polarization and relaxation are not given. It is not clear  whether in these cases Jonscher dielectric polarization and relaxation also play role or not? Due to the heterogeneity of the system Jonscher dielectric polarization and relaxation can also play a role.

These phenomena should be evaluated and, if necessary, taken into account by the determining the mechanism leakage behavior of ferroelectric layer sandwich structures at high temperature

Reviewer 3 Report

The manuscript presents how to  leakage behavior at high temperature via engineering of ferroelectric sandwich structures. Only current-voltage dependences are presented,while in any paper  dealing with ferroelctricity the P(E) curves are presented. The authors must todo the same thinkand correlate their current volatge results with P(E).

Round 2

Reviewer 3 Report

The authors have responded  wellto the reviewer's comments.